# What influences slum residents' choices of healthcare providers for common illnesses? Findings of a Discrete Choice Experiment in Ibadan, Nigeria

**Olufunke Fayehun**[1]*, **Jason Madan**[2], **Abiola Oladejo**[1], **Omobowale Oni**[3], **Eme Owoaje**[4], **Motunrayo Ajisola**[1], **Richard Lilford**[5], **Akinyinka Omigbodun**[6], **Improving Health in Slums Collaborative**[¶]

1 Department of Sociology, University of Ibadan, Ibadan, Nigeria, 2 Warwick Medical School, University of Warwick, Warwick, United Kingdom, 3 Department of Agricultural Economics, University of Ibadan, Ibadan, Nigeria, 4 Department of Community Medicine, University of Ibadan, Ibadan, Nigeria, 5 Institute of Applied Health Research, University of Birmingham, Birmingham, United Kingdom, 6 Department of Obstetrics & Gynecology, University of Ibadan, Ibadan, Nigeria

¶ Membership of the Improving Health in Slums Collaborative is provided in the Acknowledgments.
* cl_funke@yahoo.com

**Data Availability Statement:** Data cannot be shared publicly because of confidentiality. Data are available from the University of Warwick

## Abstract

Urban slum residents have access to a broad range of facilities of varying quality. The choices they make can significantly influence their health outcomes. Discrete Choice Experiments (DCEs) are a widely-used health economic methodology for understanding how individuals make trade-offs between attributes of goods or services when choosing between them. We carried out a DCE to understand these trade-offs for residents of an urban slum in Ibadan, Nigeria. We conducted 48 in-depth interviews with slum residents to identify key attributes influencing their decision to access health care. We also developed three symptom scenarios worded to be consistent with, but not pathegonian of, malaria, cholera, and depression. This led to the design of a DCE involving eight attributes with 2–4 levels for each. A D-efficient design was created, and data was collected from 557 residents between May 2021 and July 2021. Conditional-logit models were fitted to these data initially. Mixed logit and latent class models were also fitted to explore preference heterogeneity. Conditional logit results suggested a substantial Willingness-to-pay (WTP) for attributes associated with quality. WTP estimates across scenarios 1/2/3 were N5282 / N6080 / N3715 for the government over private ownership, N2599 / N5827 / N2020 for seeing a doctor rather than an informal provider and N2196 / N5421 /N4987 for full drug availability over none. Mixed logit and latent class models indicated considerable preference heterogeneity, with the latter suggesting a substantial minority valuing private over government facilities. Higher income and educational attainment were predictive of membership of this minority. Our study suggests that slum residents value and are willing to pay for high-quality care regarding staff qualifications and drug availability. It further suggests substantial variation in the perception of private providers. Therefore, improved access to government facilities and

Institutional Data Access / Ethics Committee (contact via Dr Samuel Watson) for researchers who meet the criteria for access to confidential data. The data underlying the results presented in the study are available from (The University of Warwick and https://warwick.ac.uk/fac/sci/med/about/centres/wcfgh/slums/datasets/).

**Funding:** This work was supported by funded by the National Institute for Health Research (NIHR) Global Health Research Unit on Improving Health in Slums (16/136/87 to RL, AO, EO, OF, JM) and Global Surgery (16/136/79 to RL) using UK aid from the UK Government to support global health research. The funders had no role in study design, data collection and analysis, decision to publish, or preparation of the manuscript.

**Competing interests:** The authors have declared that no competing interests exist.

initiatives to improve the quality of private providers are complementary strategies for improving overall care received.

## Introduction

Progress in treating general symptoms and several alternative approaches have made medical treatment selection a highly complex process and increasingly relevant from patients' perspectives [1–3]. Considerations on medical treatment selection include patient-physician interaction, healthcare service delivery, the concept of appropriate treatments, goal setting, and outcome measures [4]. In addition, preferences are usually personal, cultural and specific to various diseases and symptoms [5]. For example, patients may choose to travel further and pay more to see a doctor than a nurse or access a better-stocked medicine dispensary.

Understanding patient preferences is vital to designing effective healthcare policy, health systems and how service users will respond to proposed changes. The discrete choice experiment (DCE) is the preferred method for understanding patients' choices when presented with hypothetical situations competing for multifaceted interventions or service options [6]. Various alternatives in the scenarios are expressed in terms of characteristics (attributes) and associated levels, and respondents select preferred option for each set of choices. Findings from DCE methodology studies [6, 7] in healthcare have provided insight into the importance of the attributes, the relative value of the attributes and the degree to which individuals were prepared to trade between attributes or willingness to pay (WTP) to benefit or minimise harm.

The DCE application for patient preference assessment is particularly relevant in a decision scenario in which different treatment approaches are weighed between the perceived benefit and possible harm impacting the quality of health [2, 3]. This methodology can explain the equilibrium between the harms and benefits of treatment and is useful in most preferential clinical choice scenarios [8]. Therefore, the knowledge of patient needs can translate to the effective use of healthcare services and increase patient satisfaction and adherence to treatments, resulting in better outcomes [9, 10].

In low- and middle-income countries (LMICs), healthcare utilisation is closely associated with general symptoms of malaria, diarrhoea/cholera, and depression [11–13]. Malaria symptoms are non-specific, which makes it similar to other viral illnesses [14]. They include headache, tiredness, abdominal discomfort, and muscle and joint aches [14]. Approximately 241 million malaria incidents happened globally in 2020, with Nigeria accounting for about 27% of the global burden [15].

Cholera, an acute diarrhoeal infection, is a disease associated with the intake of contaminated food or water [16]. Symptoms include severe watery diarrhoea and stomach cramps with periodic vomiting. It is usually the sign of a wide range of bacterial, viral and parasite organisms that can cause digestive tract infection. It takes twelve hours to five days for an individual to show symptoms after consuming contaminated food or water [16]. Researchers estimate that there are 1.3 million to 4 million incidences of cholera, out of which 21 thousand to 143 thousand deaths have been recorded [17].

The pressure of stress and other mental health disorder is increasing worldwide. Depression is a prevalent mental illness affecting over 264 million persons worldwide [18]. Recurrent depressive disorder involves multiple episodes with individuals experiencing loss of interest and pleasure, depressed mood and decreased strength, resulting in a reduction in usual behaviour for at least two weeks [19]. There are also signs of fear, disturbed sleep and appetite, thoughts of culpability or low self-esteem [19]. About 800 thousand people die annually from suicide, with most victims between 15–29 years [19]. While well-known and effective

treatments are available for mental illnesses, about 86% of people in low- and middle-income countries are not treated for their condition [20].

Within the Nigerian context, the choice of treatment preference in urban slums became apparent from our multi-country study on improving health in slums that explores health service access and use among people who live in slum areas in four countries: Nigeria, Kenya, Pakistan, and Bangladesh [21–23]. In Nigeria, the use of formal and informal healthcare facilities varied not only by slum type but other individual characteristics. For example, in a study on contextual exploration of health use in urban slums of Nigeria, "slum residents were more likely to use formal healthcare facilities for generalised pain/other complaints, and for maternal and perinatal conditions than for communicable diseases (OR = 0.50, 95% CI 0.34 to 0.72) and non-communicable diseases (OR = 0.61, 95% CI 0.41 to 0.91). However, the differences in the odds of using formal health care facility by the predisposing factors, age, gender and marital status were not so distinct in this study. In addition, enabling resources such as employment and health insurance coverage showed different odds of using formal health care facilities" [23] pp.8. It, therefore, becomes pertinent that we study the treatment preference for commonly identified illnesses in Nigeria, such as malaria, diarrhoea/cholera and depression [15, 24–26].

This study aims to quantify the influence of service provider characteristics on respondents' choice of healthcare provider in possible cases of general symptoms associated with malaria, diarrhoea/cholera and depression.

## Materials and methods

### Study setting

This study was conducted in an urban indigenous slum in Ibadan, Nigeria, with an approximate population of 5,500 people. This is one of the seven urban slums in the NIHR Global Health Research Unit on Improving Health in Slums Project [21]. The description of this slum is detailed in other studies [21, 23]. The study population were adults aged 18 years and above.

### Ethics approval and consent to participate

Ethical approvals were obtained from the University of Warwick Biomedical and Scientific Research Ethics Committee (REGO-201702093-AM03) and Oyo State Ethics Review Committee (AD 13/479/1793A). In addition, we obtained verbal consent from each of the study participants.

### DCE research design

We adopted both qualitative (in-depth interviews) and quantitative (DCE) methods. Our condition of interest was common illnesses (malaria, diarrhoea/cholera, and depression).

### In-depth interviews and qualitative research

Telephone in-depth interviews (IDIs) were adopted because of the COVID-19 pandemic. It was used to (i) identify the symptom(s) profile that were used to finalise the scenario(s) for use in the DCE (ii) elicit attributes that participants use to make decisions about healthcare providers; (iii) determine how to present the attributes and levels pictorially or otherwise.

### Attributes identification

The DCE attribute development was informed by the secondary analysis of the responses on the choice of healthcare providers in the cross-sectional survey of the slum health project, and

a series of telephone interviews which aimed to identify the symptom(s) profile that were used to finalise the scenario(s) in the DCE, and elicit attributes influencing choice of healthcare provider.

Primary data was collected from 48 in-depth interviews (IDI). The IDI participants were purposively selected to reflect gender (male and female) and age (young [18–35 years] and old [36+ years]) characteristics of the slum residents. Telephone numbers were gotten from stakeholders who reside within the community and meet the eligibility criteria. Consent was obtained twice. After explaining the study to the potential participants, consent was obtained to contact them and phone number was collected. Consent to participate and have participants' responses recorded was also obtained before the interview commenced. During the telephone interviews, the participants were presented with the three scenarios. Through guided discussion, we established whether participants felt it would be appropriate to use these scenarios to elicit participant choices, and if any changes were required to them before they can be used. Based on the interviews and secondary data analysis findings, we established the scenario (s) to include in the DCE and their description (S1 Appendix).

The participants also identified a range of service providers and attributes that influence their choice of healthcare provider. The treatment cost was a significant influence on choice for many respondents.

*'The reason is that the cost of treatment at [Health Facility A] will be different from the [Health Facility B].'* (Male/Young/Yoruba)

*'I choose the place because I don't have much money. I will only pay a token amount there.'* (Female/Old/Hausa)

The proximity of the facility was seen as a deciding factor because the closer the service provider was, the higher the possibility of accessing health care in that facility.

*'. . .We can only go to [Health Facility A]; we cannot go to that [Health Facility B] because it is far from our area.'* (Male/Old/Yoruba)

The perceived quality of training and presence of qualified healthcare provider was highly considered in selecting a health facility,

*'You know that when you go to [Health Facility A], you may have just one doctor and others would be nurses. But if you go to [Health Facility B], they may have up to ten doctors. That's why we go there but they used to think we go there because of cheap service.'* (Female/Old/Hausa)

*'The doctor checks upon me, asks about my welfare by calling frequently. So, that gives me the grace of telling him what is wrong with me once I see his call even when I'm still thinking of calling him or going to see him. So, once I tell him he asks me to come at an appointed time; and the drugs he gives me works for me.'* (Male/Old/Yoruba)

The availability of medications was perceived as a challenge for most of the respondents. They opined that their choice of healthcare provider was sometimes determined by the availability of drugs and other services.

*'We don't get everything we need when we visit the [Health Facility A]. . . .We have to visit the [Health Facility B].'* (Male/Young/Hausa)

The type of ownership was also identified as a factor determining choice. Some participants explained it in terms of state and private ownership, which they believe determines the quality of care they would get.

*I visit there because it is government owned. I know I will be given proper care there.'* (Male/Old/Hausa)

*'I know that since it is state owned it will still be better'* (Male/Old/Yoruba)

Other factors arising from the discussion around the choice of healthcare provider included minimal drug prescription, waiting time, attachment and recommendation to a particular service provider. However, these factors were not explained as having a significant influence on participants' choice of service provider, hence their exclusion as attributes for the DCE.

## Choice of attributes and levels

Table 1 gives the list of attributes and their plausible levels that were chosen for the DCE, based on the insights from the qualitative phase. The final DCE consisted of two unlabeled alternatives and a neither option. Unlabeled alternatives were: health facility A and health facility B. The attributes used to describe the alternatives included the travel time between the participant's residence and the facility, type of ownership, medication or drug availability, privacy, gender choice, type of facility, staff and treatment cost.

**Table 1.  Attributes and levels chosen for the discrete choice experiment.**

| | Attributes | Levels |
|---|---|---|
| 1 | Treatment Cost (includes consultations, investigations, drugs) | N500 |
| | | N1,000 |
| | | N2,000 |
| | | N5,000 |
| 2 | Travel Time | 15mins |
| | | 30mins |
| | | 45mins |
| | | 1hr |
| 3 | Type of Ownership | Public |
| | | Private |
| | | Community-based management |
| 4 | Medication/Drug availability | No medication |
| | | Partially stocked dispensary |
| | | Fully stocked pharmacy |
| 5 | Confidentiality | Private consultation |
| | | Shared consultation |
| 6 | Gender Preference | Choice of provider gender, |
| | | No choice of provider gender |
| 7 | Admission Facility | Out-patient services |
| | | Day care services |
| | | In-patient services |
| 8 | Qualified trained staff | Patient Medicine Vendor |
| | | CHO/ CHEW/ Midwife/ Nurse |
| | | Medical doctor/ Pharmacist/ Medical Lab technologist |

### DCE survey development

**Participants and sampling.**   The sample frame of 1899 households whose adult members indicated that they would be happy to be re-contacted during the NIHR study [21, 27] was used for the individual DCE survey. This ensured a valid sampling frame and detailed demographic and socio-economic data on the participants to supplement the analyses. Slum residents who were not 18+ years and/or part of the sample frame were excluded from the study.

Given the lack of consensus on the minimum sample size for discrete choice experiments [28, 29], we determined a minimum sample size using the rule of thumb developed by Johnson and Orne [30]. This gave a sample size of 135 participants. This would be the minimum sample required to estimate population preferences, but a significantly larger sample would improve precision and understanding of heterogeneity. Therefore, to estimate population preferences and participant preference heterogeneity, with as much precision as possible, we set our target sample size significantly higher than the minimum required, at 726 respondents. Fig 1 shows the design and conduct of the study (Fig 1).

**Experimental design.**   A d-efficient design was created using the software NGENE 1.3 [31] in order to minimise the number of tasks required from each participant. This design led to 9 blocks of 8 choice tasks each involving 2 hypothetical health care facilities (HCFs) and an opt-out choice if the respondent would choose self-care over either of the two available HCFs. Each hypothetical HCF is defined as some combination of levels for each of the 8 attributes. To check the validity of responses, we added an additional choice task to each block where we pre-specified the levels for each hypothetical HCF so that one would be expected to always be preferred a priori. Each respondent completed 1 block per scenario; they were asked to complete 27 choice tasks which maximised the statistical efficiency of the design, i.e., provide the most information on the values of the parameters in the statistical model. We ensured that the design was orthogonal (the variation of levels across attributes is strictly independent).

### Testing the Discrete Choice Experiment (DCE) survey

The instrument was piloted on 70 individuals from a similar community to the study community to test the validity of the design and comprehension of the choice sets. Responses from this pre-pilot were used to refine the survey. Participants were 18+ years and residents of the community.

### DCE survey

Surveys were conducted on Android tablet devices using the Open Data Kit software. We followed the same data collection and storage protocol as the Improving Health in Slums project [21]. Data collection was done between May 2021 and July 2021. The data was checked and cleaned and a final anonymised, individual level data set including demographic (including age, education, and household composition) and socioeconomic (including income and household wealth) variables was derived.

### Statistical analysis

We fitted a range of statistical models to response data. We fitted multinomial regression models and then estimated more complex models that allowed for respondent heterogeneity and violation of the Independence of Irrelevant Alternatives axiom–namely, mixed logit and (2 and 3 class) latent class models. We assumed a linear relationship between cost and utility, allowing us to convert other attribute utilities into the willingness to pay (WTP) for the desired attribute. Model fit was assessed by calculating adjusted rho-squared statistics and by using the

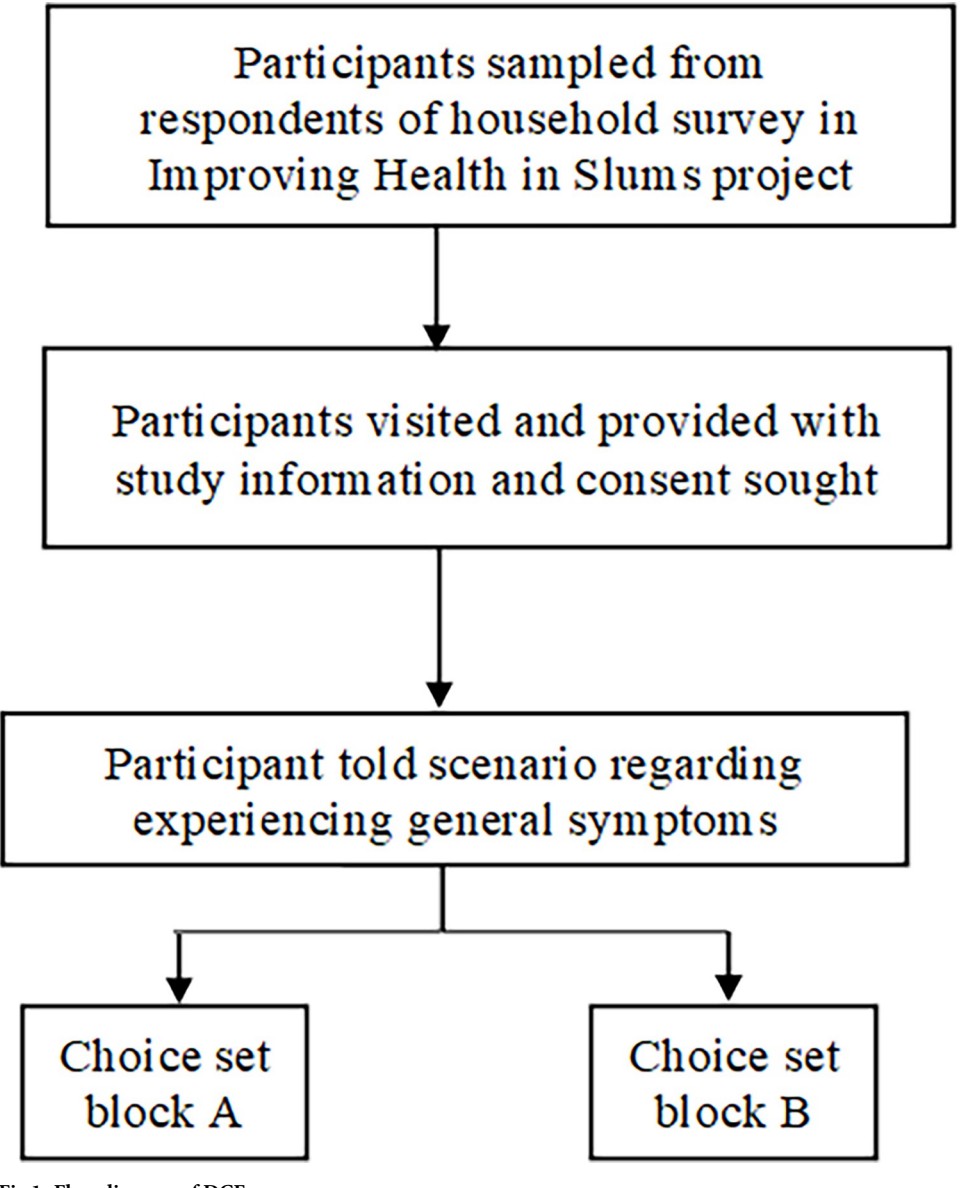

**Fig 1. Flow diagram of DCE.**

Akaike Information Criterion (AIC), where a lower AIC indicates a more parsimonious model [32]. All model fitting and statistical analysis was carried out in version 4.0.4 of the statistical software R, using the package Apollo designed specifically for choice model estimation [33].

## Results

### Description of sample

Five hundred and fifty-five people responded to the DCE. This was substantially below the target sample of seven hundred and twenty-six because of COVID-19 and associated restrictions which impeded data collection during the study period. Although this was a smaller sample size than planned, it was still above the minimum sample size of 135. Table 2 describes the sociodemographic profile of respondents. The study respondents were predominately female

**Table 2. Sociodemographic characteristics of respondents.**

| | |
|---|---|
| **Total Sample** | 557 |
| **Female** | 364 (65%) |
| **Income > N25000 / month** | 270 (48%) |
| **No secondary education** | 277 (49%) |
| **Mean Age** | 49 years (IQR 35–64 years) |
| **Household size 5+** | 204 (37%) |

(65%) and evenly split in terms of earning above or below N25000 (approx. USD60) per month and whether they had any education past primary level schooling. Of the nine questions added as a check of process validity, the percentage answered in line with our a priori expectations ranged from 53% - 85%, with five answering in line with expectations over 60% of the time.

## Overview of model results

Table 3 summarises the performance of each of the five models fitted to DCE response data for each of the three scenarios. All models except for Conditional Logit allow for preference heterogeneity among respondents. Across all scenarios, Conditional Logit displayed poorer model fit, suggesting that there is substantial heterogeneity in the preferences of our respondents. The mixed logit model performed well across all scenarios, and including specific interaction terms for education and income further improved model fit for scenarios 1 and 2, suggesting these factors influence preference heterogeneity for those scenarios. Increasing the number of classes in the Latent Class Model from two to three worsened model fit, suggesting our data best supports two subgroups with differing preferences.

We present further details of results for these models below. Results are presented in terms of both level regression coefficients (βs) and WTP in Naira.

## Conditional logit model

Table 4 presents the conditional logit model results for each of the scenarios. The analysis reported that the level 'outpatient' for attribute 'facility' and the level 'nurse' for attribute 'staff qualification' did not influence choices in any models, and no results are reported for these levels. There was a strong preference for the unlabelled HCF alternatives over self-care for all scenarios. There are striking similarities across scenarios, such as the preference for lower-cost over higher-cost facilities, government over privately-owned facilities, doctors over health care workers (HCWs) without a formal qualification, and full drug availability over none. All of

**Table 3. Model fit statistics.**

| | Scenario 1 | | Scenario 2 | | Scenario 3 | |
|---|---|---|---|---|---|---|
| | Adjusted rho^2 | AIC | Adjusted rho^2 | AIC | Adjusted rho^2 | AIC |
| Conditional Logit | 0.3825 | 6801.55 | 0.3061 | 7643.32 | 0.3414 | 7278.28 |
| Mixed Logit | 0.5039 | 5464.59 | 0.4396 | 6173.07 | 0.4857 | 5711.14 |
| Latent Class Model (2-class) | 0.4723 | 5797.22 | 0.4104 | 6482.43 | 0.4580 | 6011.96 |
| Latent Class Model (3-class) | 0.4637 | 5907.26 | 0.4068 | 6534.46 | 0.4495 | 6063.51 |
| Mixed Logit with interactions | 0.4890 | 5629 | 0.4096 | 6503 | 0.4519 | 6037 |

AIC–Akaike Information Criterion (lower value indicates more parsimonious fit)

**Table 4. Conditional logit model results for each scenario.**

| Attribute | Level | Reference | Conditional Logit | | | | | |
|---|---|---|---|---|---|---|---|---|
| | | | Scenario 1 | | Scenario 2 | | Scenario 3 | |
| | | | B (Mean and 95% CI) | WTP (N) | B (Mean and 95% CI) | WTP (N) | B (Mean and 95% CI) | WTP (N) |
| **Visit an HCF?** | Reference HCF | Self care | 4.06 (3.53,4.59) | | 3.32 (2.87,3.77) | | 3.47 (2.92,4.03) | |
| **Option B bias** | Option B | Option A | 0.02 (-0.04,0.09) | | -0.01 (-0.05,0.04) | | -0.11 (-0.15,-0.05) | |
| **Travel Time** | 30-minute increase | | -0.40 (-0.49,-0.29) | -2953 | -0.24 (-0.36,-0.13) | -3978 | -0.06 (-0.16,0.03) | -560 |
| **Facility Ownership** | Community | Government | -0.08 (-0.2,0.03) | -610 | 0.06 (-0.09,0.22) | 1175 | -0.02 (-0.15,0.1) | -178 |
| | Private | Government | -0.69 (-0.84,-0.54) | -5282 | -0.36 (-0.51,-0.22) | -6080 | -0.4 (-0.54,-0.25) | -3715 |
| **Drug availability** | Partial | None | 0.09 (0,0.18) | 725 | 0.12 (0.03,0.22) | 2237 | 0.16 (0.05,0.27) | 1593 |
| | Full | None | 0.29 (0.14,0.43) | 2196 | 0.31 (0.15,0.48) | 5421 | 0.51 (0.41,0.62) | 4987 |
| **Privacy** | Yes | No | 0.03 (-0.04,0.1) | 282 | -0.01 (-0.11,0.1) | -13 | 0.02 (-0.04,0.07) | 191 |
| **Gender Choice** | Yes | No | -0.05 (-0.14,0.03) | -344 | -0.09 (-0.16,-0.02) | -1477 | -0.09 (-0.17,-0.01) | -820 |
| **Facility** | Inpatient | Daycare | 0.0 (-0.08,0.09) | 45 | 0.08 (-0.04,0.19) | 1386 | -0.06 (-0.17,0.05) | -820 |
| **Staff Qualification** | Doctor | No formal qualification | 0.34 (0.22,0.45) | 2599 | 0.34 (0.23,0.44) | 5827 | 0.21 (0.1,0.31) | 2020 |
| **Cost** | N1000 increase | | -0.14 (-0.16,-0.11) | | -0.06 (-0.09,-0.03) | | -0.11 (-0.15,-0.07) | |

HCF–Health Care Facility; CI–Confidence Interval; WTP–Willingness to Pay

these preferences are significant at the 5% level across scenarios. The results also show some differences between scenarios; most noticeably, travel time is a significant decision attribute for scenarios one and two, but not scenario three. For scenarios one and two, government rather than private ownership was a stronger driver of preference than the next strongest factor, full drug availability. While these remained significant choice drivers of preference for scenario three, their order of importance is reversed. Privacy and gender choice consistently had minimal impact on healthcare provider choice across all three scenarios.

## Mixed logit model

Table 5 presents results from the mixed logit model, which allows for random heterogeneity in respondents' preferences. The results presented reflect the preferences of the 'typical' respondent since mixed logit models allow for individuals to have their own preferences. Broadly speaking, the typical respondent exhibits preferences consistent with, but stronger than, those estimated by the conditional logit model. For example, as in the conditional logit model, the reference HCF is preferred to self-care, but the β coefficients for this are considerably higher across all scenarios. The same is true for the other statistically significant attribute/levels such as government vs private ownership, full drug availability, and interaction with a doctor rather than an unqualified healthcare worker.

## Latent class model

The latent class model results presented are for the 2-class model, which was preferred to the 3-class model (S1 Table) due to its lower AIC. While improving overall model fit, the mixed logit results do not substantially alter the population-level predictions of utility and WTP. However, the latent class model provides potential insights into this heterogeneity. The classification model (Table 6) suggests that income and education are important influences on class membership, with one class more likely to include those with lower income and educational attainment than the other. However, income is not a statistically significant predictor of class

**Table 5. Mixed logit model results for each scenario.**

| Attribute | Level | Reference | Mixed Logit | | | | | |
| --- | --- | --- | --- | --- | --- | --- | --- | --- |
| | | | Scenario 1 | | Scenario 2 | | Scenario 3 | |
| | | | B (Mean and 95% CI) | WTP (N) | B (Mean and 95% CI) | WTP (N) | B (Mean and 95% CI) | WTP (N) |
| **Visit an HCF?** | Reference HCF | Self care | 8.03 (4.98,8.78) | | 10.38 (8.43,12.32) | | 7.95 (7.85,8.04) | |
| **Option B bias** | Option B | Option A | 0.02 (-0.04,0.08) | | -0.06 (-0.15,0.02) | | -0.17 (-0.29,-0.06) | |
| **Travel Time** | 30-minute increase | | -0.87 (-1.17,-0.56) | -2746 | -0.45 (-0.67,-0.22) | -3383 | -0.27 (-0.54,0.0) | -1002 |
| **Facility Ownership** | Community | Government | -0.14 (-0.4,0.13) | -415 | 0.09 (-0.27,0.44) | 673 | 0.07 (-0.3,0.43) | 280 |
| | Private | Government | -1.77 (-2.41,-1.17) | -5684 | -0.89 (-1.27,-0.53) | -6782 | -1.08 (-1.56,-0.58) | -3983 |
| **Drug availability** | Partial | None | 0.32 (0.08,0.57) | 1057 | 0.36 (0.17,0.55) | 2767 | 0.73 (0.52,0.95) | 2765 |
| | Full | None | 1.1 (0.77,1.44) | 3558 | 0.87 (0.54,1.19) | 6643 | 1.03 (0.77,1.29) | 3983 |
| **Privacy** | Yes | No | 0.13 (-0.01,0.28) | 434 | 0.06 (-0.07,0.18) | 459 | 0.06 (-0.16,0.28) | 235 |
| **Gender Choice** | Yes | No | -0.01 (-0.06,0.03) | -31 | -0.21 (-0.34,-0.09) | -1596 | -0.27 (-0.46,-0.08) | -1009 |
| **Facility** | Inpatient | Daycare | 0.1 (-0.12,0.31) | 337 | 0.12 (-0.19,0.42) | 944 | -0.01 (-0.06,0.03) | -34 |
| **Staff Qualification** | Doctor | No formal qualification | 0.9 (0.59,1.22) | 2928 | 0.83 (0.59,1.08) | 6376 | 0.69 (0.45,0.93) | 2605 |
| **Cost** | N1000 increase | | -0.31 (-0.4,-0.24) | | -0.14 (-0.22,-0.05) | | -0.27 (-0.37,-0.17) | |

HCF–Health Care Facility; CI–Confidence Interval; WTP–Willingness to Pay

membership, and education is only a statistically significant predictor for scenarios one and two. Nevertheless, we categorise the income classes as low and high socioeconomic status (SES) to aid the explanation.

Some differences in preferences emerge between the low and high SES classes (Table 6). The population preference for the government over private facilities is found in the low SES class.

For scenarios 2 and 3, respondents in the high-SES have the opposite preference, expressing a high WTP for private over government facilities (for scenario 1 the relationship between cost and utility is counter-intuitively positive, albeit non-significant, preventing the estimation of meaningful WTPs). The high SES aversion to government-owned facilities extends to a preference for community-owned facilities.

The low SES population has non-significant preferences between community-owned and government-owned facilities across scenarios. The high-SES population also appear more willing to pay to reduce travel time. The proportion of respondents allocated to the low-SES class is 82.5%, 57% and 37% for the three scenarios. The high proportion for scenario 1 suggests

**Table 6. Predictive impact of socioeconomic indicators on class membership in 2-class latent class model.**

| | | Scenario 1 | Scenario 2 | Scenario 3 |
| --- | --- | --- | --- | --- |
| **% in class A** | | **83%** | **57%** | **37%** |
| Influence of socioeconomic factors on odds of membership of class A vs class B (Odds Ratio, mean and 95% C.I.) | High Income | 0.75 (0.45,1.25) | 0.74 (0.51,1.09) | 0.79 (0.54,1.17) |
| | High Education | 0.58 (0.34,0.97) | 0.61 (0.42,0.89) | 0.78 (0.53,1.17) |
| | Female | 1.33 (0.80,2.19) | 1.17 (0.78,1.74) | 0.96 (0.61,1.50) |
| | Household size 5 + | 0.95 (0.59.1.32) | 0.78 (0.53,1.14) | 0.94 (0.67,1.32) |

that preferences are less heterogeneous in this scenario and that the results for the high-SES class need to be interpreted with caution.

## Mixed logit model with interaction terms

Given the relationship between income/education and utility functions suggested by the latent class models, we refitted the mixed logistic regression model with interaction terms between attribute-level coefficients and these SES variables. The resulting coefficients from these models for the three scenarios are shown in Tables 7 and 8.

There are significant interactions between high income and education and reduced preference for government-owned facilities across scenarios, consistent with the findings of the latent class models. The β-coefficients reflecting the relative utility of government vs private ownership are 0.3, 0.46 and 0.45 for scenarios one, two and three among those with low income and low educational background (significantly above zero at the 5% level for scenarios 2 and 3). They remain positive for those with low income but high education (0.07, 0.16, and 0.15 across scenarios) but are no longer significant for any scenario. For those with low education but high income, the β-coefficients are all below zero (-0.8, -0.4, and -0.01), although this is only statistically significant for scenario one.

## Discussion

This study advances the understanding of the choice of healthcare providers in the urban slums of developing countries. To the best of our knowledge, this is the first study to estimate slum residents' preferences and trade-offs when deciding if and when to seek care in Nigeria and West Africa. A previous study in Malawi, East Africa, focused on caregivers' preference for under-five child healthcare services in urban slums [34].

The results highlight the importance of health service provider characteristics on the choice made by slum residents. This affirms findings from previous studies [34, 35] that respondents are willing to pay substantial sums to access healthcare with desirable attributes rather than prioritising cost and convenience. Moreover, the result aligns with our previous findings regarding using facilities in slums, where people frequently bypass nearby facilities to obtain care from the government hospital outpatient department [27]. For example, a related DCE conducted in Malawi showed that respondents would pay more if the facility had medicine and supplies and could thoroughly examine their children [34]. Another similar study in Uganda reported that participants were willing to pay more for health insurance covering certain health conditions and treatments [36].

High values are placed on facility ownership, with respondents preferring government over private facilities. We think the most plausible reason for this is the perceived quality of care and the availability of trained personnel. As found in another study, appropriateness of care, availability of a physician or nurse, and drugs significantly influence preferences [35, 37, 38]. In addition, highly valued attributes are staff qualifications (doctor preferred to informal provider) and fully stocked dispensaries. On the other hand, privacy and staff gender were not highly valued as attributes of choice in this study context. This suggests that respondents focused primarily on outcomes rather than experience when considering their facility choice [5].

There was evidence of preference heterogeneity within our sample. This was most evident in the value placed on facility ownership. Latent class analysis suggests that a significant minority have divergent values, strongly preferring privately-owned facilities [39]. Analysis of latent class membership predictors suggests that those choosing private to government facilities are more likely to be (relatively) well-educated and on higher incomes. Similar results were

**Table 7. Latent class (2-class) model results for each scenario.**

| Attribute | Level | Reference | Scenario 1 | | | | Scenario 2 | | | | Scenario 3 | | | |
|---|---|---|---|---|---|---|---|---|---|---|---|---|---|---|
| | | | Class A | | Class B | | Class A | | Class B | | Class A | | Class B | |
| | | | B (Mean & 95% CI) | WTP (N) | B (Mean & 95% CI) | WTP (N) | B (Mean & 95% CI) | WTP (N) | B (Mean & 95% CI) | WTP (N) | B (Mean & 95% CI) | WTP (N) | B (Mean & 95% CI) | WTP (N) |
| Visit an HCF? | Reference HCF | Self care | 5.81 (4.91,6.71) | NA | 1.95 (0.59,3.31) | NA | 3.66 (2.88,4.44) | NA | 4.32 (3.2,5.43) | NA | 3.38 (1.96,4.8) | NA | 3.48 (2.8,4.16) | NA |
| Option B bias | Option B | Option A | 0.03 (-0.05,0.1) | | | NA | 0.02 (-0.04,0.08) | NA | | NA | -0.15 (-0.24,-0.07) | | | |
| Travel Time | 30-minute increase | | -0.39 (-0.51,-0.26) | -1618 | -0.87 (-1.47,-0.26) | NA | 0.25 (0.08,0.42) | 1194 | -1.24 (-1.77,-0.69) | -5800 | 1.08 (0.57,1.58) | 4636 | -0.48 (-0.63,-0.33) | -2008 |
| Facility Ownership | Community | Government | -0.02 (-0.17,0.13) | -68 | 0.15 (-0.33,0.63) | NA | 0.08 (-0.15,0.3) | 374 | 0.75 (0.11,1.38) | 3569 | -0.39 (-0.76,-0.02) | -1618 | 0.39 (0.12,0.65) | 1686 |
| | Private | Government | -1.24 (-1.43,-1.08) | -5243 | 1.58 (0.9,2.25) | NA | -1.31 (-1.77,-0.87) | -6086 | 0.88 (0.5,1.26) | 4194 | -3 (-3.51,-2.21) | -12188 | 0.55 (0.32,0.79) | 2399 |
| Drug availability | Partial | None | 0.18 (0.07,0.29) | 779 | -0.6 (-1.05,-0.15) | NA | 0.4 (0.26,0.54) | 1902 | -0.33 (-0.6,-0.05) | -1514 | 0.63 (0.25,1) | 2711 | 0.2 (0.08,0.32) | 873 |
| | Full | None | 0.39 (0.22,0.58) | 1690 | -0.17 (-0.76,0.39) | NA | 0.53 (0.33,0.73) | 2522 | -0.53 (-1.14,0.07) | -2458 | 0.97 (0.68,1.27) | 4194 | 0.51 (0.34,0.69) | 2227 |
| Privacy | Yes | No | -0.02 (-0.12,0.09) | -43 | 0.31 (0.1,0.52) | NA | 0.08 (-0.11,0.25) | 381 | -0.27 (-0.58,0.03) | -1246 | 0.4 (0.14,0.66) | 1740 | -0.08 (-0.16,0.01) | -329 |
| Gender Choice | Yes | No | -0.2 (-0.3,-0.08) | -792 | 0.36 (0.03,0.7) | NA | -0.14 (-0.24,-0.04) | -623 | -0.42 (-0.62,0.2) | -1922 | -0.27 (-0.49,0.06) | -1156 | -0.11 (-0.24,0.03) | -409 |
| Facility | Inpatient | Daycare | 0.08 (-0.03,0.18) | 341 | -0.19 (-0.54,0.17) | NA | 0.13 (-0.11,0.37) | 650 | 0.39 (-0.01,0.79) | 1875 | -0.48 (-1.05,0.09) | -2038 | -0.01 (-0.24,0.21) | -40 |
| Staff Qualification | Doctor | Informal provider | 0.44 (0.29,0.6) | 1901 | 0.17 (-0.29,0.6) | NA | 0.72 (0.54,0.9) | 3428 | -0.25 (-0.65,0.17) | -1142 | 1.27 (0.54,1.99) | 5439 | 0.09 (-0.13,0.31) | 408 |
| Cost | N1000 increase | | -0.25 (-0.29,0.2) | | 0.1 (0,0.19) | NA | -0.03 (-0.09,0.04) | | -0.22 (-0.34,-0.09) | | -0.13 (-0.26,0.01) | | -0.24 (-0.31,0.17) | |

HCF–Health Care Facility; CI–Confidence Interval; WTP–Willingness to Pay

**Table 8. Mixed logit model with interactions results for each scenario.**

| | Attribute | Level | Reference | Scenario 1 | Scenario 2 | Scenario 3 |
|---|---|---|---|---|---|---|
| BASELINE COEFFICIENTS | Visit HCF | Reference HCF | Self Care | 9.46 (7.48,11.4) | 10.6 (8.04,13.3) | 18.1 (13.0,23.1) |
| | Option B bias | Option B | Option A | 0.04 (-0.1,0.27) | -0.0 (-0.1,0.04) | -0.1 (-0.3,-0.0) |
| | Travel time | 30 min inc | NA | 2.29 (1.25,3.32) | 1.35 (0.77,1.92) | 0.91 (0.31,1.51) |
| | HCF owner | Community | Private | -0.8 (-1.3,-0.4) | -0.3 (-0.6,-0.0) | -0.1 (-0.3,0.09) |
| | | Govt | Private | 0.30 (-0.1,0.77) | 0.46 (0.13,0.79) | 0.45 (0.16,0.74) |
| | Drug Stocks | Partial | None | 2.26 (1.53,3.00) | 1.35 (0.87,1.83) | 1.17 (0.61,1.73) |
| | | Full | None | 0.16 (-0.5,0.91) | 0.15 (-0.1,0.46) | 0.03 (-0.3,0.45) |
| | Privacy | Yes | No | 1.01 (0.40,1.63) | 1.04 (0.58,1.50) | 0.92 (0.49,1.35) |
| | Gender Choice | Yes | No | -0.1 (-0.6,0.25) | 0.15 (-0.2,0.58) | -0.0 (-0.3,0.22) |
| | Facility | Inpatient | Daycare | 0.16 (-0.1,0.47) | -0.0 (-0.3,0.14) | -0.3 (-0.7,0.01) |
| | Qualification | Doctor | None | -0.4 (-0.8,0.03) | -0.1 (-0.2,0.00) | -0.3 (-0.4,-0.2) |
| | Cost | N1000 inc | NA | 0.97 (-0.0,1.95) | 0.92 (0.55,1.30) | 0.57 (0.15,0.99) |
| Interaction—high income | Visit HCF | Reference HCF | Self Care | 0.12 (-0.6,0.92) | 0.02 (-0.0,0.09) | 0.04 (-0.2,0.31) |
| | Travel time | 30 min inc | NA | 0.41 (-1.1,1.97) | -1.2 (-2.9,0.38) | -0.1 (-2.7,2.38) |
| | HCF owner | Community | Private | -0.3 (-1.1,0.36) | -0.5 (-1.0,-0.0) | -0.5 (-1.0,-0.0) |
| | | Govt | Private | -0.8 (-1.8,0.08) | -0.4 (-1.0,0.13) | -0.01 (-0.5,0.32) |
| | Drug Stocks | Partial | None | -0.0 (-0.0,0.07) | -0.4 (-1.0,0.18) | -0.1 (-0.6,0.34) |
| | | Full | None | 0.04 (-0.6,0.70) | -0.3 (-0.6,0.01) | -0.01 (-0.1,0.04) |
| | Privacy | Yes | No | -0.0 (-0.2,0.16) | 0.02 (-0.1,0.23) | -0.0 (-0.3,0.15) |
| | Gender Choice | Yes | No | 0.18 (-0.4,0.83) | -0.0 (-0.0,0.04) | -0.1 (-0.6,0.36) |
| | Facility | Inpatient | Daycare | 0.39 (-0.1,0.97) | 0.05 (-0.4,0.54) | 0.09 (-0.2,0.46) |
| | Qualification | Doctor | None | 0.50 (0.08,0.91) | 0.04 (-0.0,0.15) | 0.09 (-0.4,0.63) |
| | Cost | N1000 inc | NA | -1.0 (-3.4,1.35) | -0.1 (-1.2,1.02) | -4.7 (-7.1,-2.3) |
| Interaction–high education | Visit HCF | Reference HCF | Self Care | 0.11 (-0.4,0.71) | 0.04 (-0.3,0.44) | 0.01 (-0.0,0.08) |
| | Travel time | 30 min inc | NA | -1.2 (-2.4,-0.0) | -0.5 (-1.2,0.02) | -0.4 (-1.1,0.34) |
| | HCF owner | Community | Private | 0.04 (-0.5,0.67) | -0.1 (-0.6,0.29) | -0.0 (-0.3,0.18) |
| | | Govt | Private | 0.07 (-0.4,0.63) | 0.16 (-0.3,0.69) | 0.15 (-0.2,0.54) |
| | Drug Stocks | Partial | None | -1.1 (-1.9,-0.2) | -0.6 (-1.2,-0.0) | -0.5 (-1.3,0.31) |
| | | Full | None | -0.0 (-1.2,1.06) | -0.1 (-0.4,0.17) | 0.19 (-0.0,0.43) |
| | Privacy | Yes | No | 0.41 (-0.1,0.96) | 0.11 (-0.9,1.14) | 0.38 (-0.3,1.07) |
| | Gender Choice | Yes | No | -0.1 (-0.9,0.70) | -0.0 (-0.1,0.04) | -0.0 (-0.3,0.23) |
| | Facility | Inpatient | Daycare | 0.02 (-0.1,0.16) | -0.2 (-0.6,0.10) | 0.50 (0.14,0.86) |
| | Qualification | Doctor | None | 0.12 (-0.2,0.51) | -0.0 (-0.3,0.29) | 0.15 (-0.0,0.32) |
| | Cost | N1000 inc | NA | -0.1 (-1.8,1.53) | -0.2 (-0.7,0.21) | 0.14 (-0.4,0.73) |

HCF–Health Care Facility; CI–Confidence Interval; WTP–Willingness to Pay

reported on preferences heterogeneity of health care utilisation in China [40]. A related observation from other work carried out by our Unit is that there tends to be a greater spread of attractiveness within privately-owned facilities. Together, these findings support the interpretation that most slum residents have access to low-quality private facilities that they prefer not to use but that a minority have the resources and knowledge to access private facilities they perceive as high-quality [41].

While these findings are broadly similar across scenarios, some differences exist. For example, the greater weight is given to travel time in scenarios one and two (fever and diarrhoea), while a higher proportion of those who prefer private to government-owned facilities are in

scenarios two and three. This finding is consistent with results from our previous study [23], which suggests that presenting medical complaints is a strong determinant of formal healthcare utilisation in urban slums of Nigeria.

Limitation to our study is the small sample size which led to imbalanced sampling across blocks. This is because of the restriction from the COVID-19 pandemic. This small sample size restricted how far we could explore heterogeneity and could not estimate utilities for two levels (outpatient availability and nurse-qualified staff). Our results suggest that outpatient availability is unlikely to influence choice, but further research would help establish the impact of nurse-delivered care. We also found that responses to our validity questions were not always common in line with our a priori expectations. While this might raise concerns over whether the derived results reasonably reflect the actual preferences and values of respondents, our view is that there is more likely to be a consequence of more significant variation in preferences than we anticipated when designing our validity questions. The discrepancy between our expected responses and some actual ones reflects flaws in our predictions of respondent preferences rather than any inability of respondents to make choices consistent with their preferences.

## Conclusion

Our study suggests that slum residents value and are willing to pay and travel further for high-quality care regarding staff qualifications and drug availability. It further suggests substantial variation in the perception of private providers. Those who are better-off and highly educated tend to view private providers more favorably, suggesting they have the awareness and means to select high-quality private providers. Improved access to government facilities and initiatives to improve the quality of private providers are complementary strategies for improving overall care received.

## Supporting information

**S1 Appendix. Three symptom scenarios.**
(DOCX)

**S1 Table. Results of the mixed logit and latent class with 3 models for each of the scenarios.**
(DOCX)

## Acknowledgments

Members of the Improving Health in Slums Collaborative are Pauline Bakibinga, Caroline Kabaria, Ziraba Kasiira, Peter Kibe, Catherine Kyobutungi, Nelson Mbaya, Blessing Mberu, Shukri Mohammed, Anne Njeri, Lyagamula Kisia, Iqbal Azam, Romaina Iqbal, Ahsana Nazish, Narjis Rizvi, Kehkashan Azeem, Syed Shifat Ahmed, Omar Rahman, Rita Yusuf, Nazratun Choudhury, Oladoyin Odubanjo, Motunrayo Ajisola, Olufunke Fayehun, Akinyinka Omigbodun, Mary Osuh, Eme Owoaje, Olalekan Taiwo, Richard Lilford, Jo Sartori, Sam Watson, Peter Diggle, Navneet Aujla, Yen-Fu Chen, Paramjit Gill, Frances Griffiths, Bronwyn Harris, Jason Madan, Oyinlola Oyebode, Joao Porto De Albuquerque, Simon Smith, Ola Uthman, Ria Wilson, Godwin Yeboah, Grant Tregonning, Ji-Eun Park

## Author Contributions

**Conceptualization:** Olufunke Fayehun, Jason Madan, Eme Owoaje, Motunrayo Ajisola, Richard Lilford, Akinyinka Omigbodun.

**Formal analysis:** Olufunke Fayehun, Jason Madan, Abiola Oladejo, Omobowale Oni, Motunrayo Ajisola.

**Funding acquisition:** Olufunke Fayehun, Jason Madan, Eme Owoaje, Richard Lilford, Akinyinka Omigbodun.

**Methodology:** Olufunke Fayehun, Jason Madan, Abiola Oladejo, Omobowale Oni, Eme Owoaje, Motunrayo Ajisola, Richard Lilford, Akinyinka Omigbodun.

**Project administration:** Olufunke Fayehun, Richard Lilford, Akinyinka Omigbodun.

**Supervision:** Akinyinka Omigbodun.

**Writing – original draft:** Olufunke Fayehun, Jason Madan, Abiola Oladejo, Omobowale Oni, Eme Owoaje, Motunrayo Ajisola, Richard Lilford, Akinyinka Omigbodun.

**Writing – review & editing:** Olufunke Fayehun, Jason Madan, Abiola Oladejo, Richard Lilford, Akinyinka Omigbodun.

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
