## [Decision Letter · Decision Letter 0]

1 Dec 2022

PGPH-D-22-01545

What influences slum residents’ choices of healthcare providers for common illnesses? Findings of a Discrete Choice Experiment in Ibadan, Nigeria

Dear Dr. Fayehun,

Thank you for submitting your manuscript to PLOS Global Public Health. After careful consideration, we feel that it has merit but does not fully meet PLOS Global Public Health’s publication criteria as it currently stands. Therefore, we invite you to submit a revised version of the manuscript that addresses the points raised during the review process. The three reviewers generally found the experimental design sound, but raised concerns regarding the clarity of presentation that I believe should be feasible to address, particularly separating the tables or shifting some findings to supplement to ensure the results that are presented are clearly explained in the text. (Please disregard the brief opinion on not liking a particular tables, but do consider how to ensure clarity of the complex results presented.)

We look forward to receiving your revised manuscript.

Kind regards,

Hannah Hogan Leslie, PhD

Academic Editor

Journal Requirements:

a) State the initials, alongside each funding source, of each author to receive each grant. For example: "This work was supported by the National Institutes of Health (####### to AM; ###### to CJ) and the National Science Foundation (###### to AM)."

2. Please update your online Competing Interests statement. If you have no competing interests to declare, please state: “The authors have declared that no competing interests exist.”

3. Please provide separate figure files in .tif or .eps format only and remove any figures embedded in your manuscript file. Please also ensure that all files are under our size limit of 10MB.

4. We have noticed that you have a list of Supporting Information legends in your manuscript. However, there are no corresponding files uploaded to the submission. Please upload them as separate files with the item type 'Supporting Information'. 

5. Please include copies of Supplementary Table 1 and Supplementary Table 2, which you refer to in your text on pages 7 and 14. Please upload them as separate files with the item type 'Supporting Information'. Please ensure that each Supporting Information file has a legend listed in the manuscript after the references list.

6. We notice that your supplementary table (Appendix 1) is included in the manuscript file. Please remove them and upload them with the file type 'Supporting Information'. Please ensure that each Supporting Information file has a legend listed in the manuscript after the references list.

Additional Editor Comments (if provided):

Reviewers' comments:

Reviewer's Responses to Questions

**Comments to the Author**

1. Does this manuscript meet PLOS Global Public Health’s publication criteria? Is the manuscript technically sound, and do the data support the conclusions? The manuscript must describe methodologically and ethically rigorous research with conclusions that are appropriately drawn based on the data presented.

Reviewer #1: Yes

Reviewer #2: Yes

Reviewer #3: Partly

2. Has the statistical analysis been performed appropriately and rigorously?

Reviewer #1: Yes

Reviewer #2: N/A

Reviewer #3: No

3. Have the authors made all data underlying the findings in their manuscript fully available (please refer to the Data Availability Statement at the start of the manuscript PDF file)?

Reviewer #1: Yes

Reviewer #2: No

Reviewer #3: No

4. Is the manuscript presented in an intelligible fashion and written in standard English?

Reviewer #1: Yes

Reviewer #2: Yes

Reviewer #3: Yes

5. Review Comments to the Author

Reviewer #1: Abstract

a. Higher income and educational attainment were predictive of membership of this minority – It would be nice to have the inferential statistical values that affirms the prediction in brackets right after the sentence.

Background

b. Researchers estimate……... Is this “1.3-4 million cholera incidence” a global or national statistic?

c. Kindly recast the Odds Ratios narrative in the introduction.

d. The gap in literature (and why DCE needs to be studied) has not been sufficiently captured.

Methods

e. The date (month and year) of data collection is not clearly stated anywhere – kindly provide dates of data collection.

f. This is one of the seven urban slums in the NIHR Global Health Research Unit on Improving Health in Slums Project [21] – Can authors provide the name of the slum please?

g. A table that presents the details of the 48 IDI participants will be desirable.

h. It is not also clear where and how the authors acquired the telephone numbers of the 48 participants for the IDI. Were the numbers obtained from a directory or database?

i. It is preferred that the IDI excerpts be moved to the results section instead of them appearing in the materials and methods section

j. It would be desirable if the methods and results were reported in past tense considering that the research is completed and under review.

Reviewer #2: The manuscript has an interesting work and is well written. Observations from the methodological part are listed below.

1. Consent: Were the telephonic interviews were recorded? If so, was verbal consent also obtained for the recording? Please mention that as well.

2. DCE research design: How were the IDI participants selected? whether the chosen participants had a history of either any of malaria, diarrhoea/cholera, or depression symptoms? Please eloquently discuss this.

3. Was there any inclusion or exclusion criteria for selecting participants for both IDI and for the DCE survey? A succinct description on this will be beneficial.

4. The first column in the fiig 1, is bit confusing can be re written as "Participants sampled from

respondents of household survey in Improving Health in Slums project"

Reviewer #3: 1. Does this manuscript meet PLOS Global Public Health’s publication criteria? Is the manuscript technically sound, and do the data support the conclusions? The manuscript must describe methodologically and ethically rigorous research with conclusions that are appropriately drawn based on the data presented.

The manuscript needs revision to meet the stringent presentation of results in a way that sound conclusions can be drawn from the results.

The manuscript is written with a lot of excitement and full of materials that are vital to communicating findings of a Discrete Choice Experiment (DCE).

What the key message of this manuscript? I fail to find that in the manuscript.

2. Has the statistical analysis been performed appropriately and rigorously?

It is one thing to generate statistical tables, but it is another thing to communicate your findings in a stepwise fashion. These are my charges to the team.

1. The sociodemographic are not well presented in a scientific fashion -I don’t like that table (opinion).

2. You tend to lump results that should be presented in a step wise or separate tables fashion for easy appreciation by readers. This can be done as follows.

a. clogit model.

b. -mlogit model for the presentation of the preferences of the respondents. Reporting of the beta coefficients.

c. -latent class analysis for the identification of the class phenotypes.

d. Predictors of the latent class membership

e. Willingness to pay analysis/ willingness to trade analysis

3. Reporting of the results text is missing the reporting of the beta-coefficients, their comparisons by level of strength and 95% confidence intervals. This is the main gist of a DCEs that you have completely at the mlogit and the wtp analysis.

3. Have the authors made all data underlying the findings in their manuscript fully available (please refer to the Data Availability Statement at the start of the manuscript PDF file)? -No

4. Is the manuscript presented in an intelligible fashion and written in standard English? _Yes

Additions:

The manuscripts cited in the manuscript are high quality and should guide the authors in generating a quality manuscript. Here is an article that I am currently reading and the other that has good content that the team can draw inspiration from. 1,2

1. Eshun-Wilson, I. et al. Preferences for COVID-19 vaccine distribution strategies in the US: A discrete choice survey. PLoS One 16, (2021).

2. Zanolini, A. et al. Understanding preferences for HIV care and treatment in Zambia: Evidence from a discrete choice experiment among patients who have been lost to follow-up. PLoS Med 15, 1–15 (2018).

6. PLOS authors have the option to publish the peer review history of their article (what does this mean?). If published, this will include your full peer review and any attached files.

**Do you want your identity to be public for this peer review?** For information about this choice, including consent withdrawal, please see our Privacy Policy.

Reviewer #1: **Yes: **Taiwo Akinyode OBEMBE

Reviewer #2: **Yes: **Nolita Dolcy Saldanha

Reviewer #3: **Yes: **Raphael Mando Onyango

---

## [Decision Letter · Decision Letter 1]

10 Feb 2023

What influences slum residents’ choices of healthcare providers for common illnesses? Findings of a Discrete Choice Experiment in Ibadan, Nigeria

PGPH-D-22-01545R1

Dear Dr Fayehun,

We are pleased to inform you that your manuscript 'What influences slum residents’ choices of healthcare providers for common illnesses? Findings of a Discrete Choice Experiment in Ibadan, Nigeria' has been provisionally accepted for publication in PLOS Global Public Health.

Best regards,

Hannah Hogan Leslie, PhD

Academic Editor

Reviewer Comments (if any, and for reference):

Reviewer's Responses to Questions

**Comments to the Author**

1. If the authors have adequately addressed your comments raised in a previous round of review and you feel that this manuscript is now acceptable for publication, you may indicate that here to bypass the “Comments to the Author” section, enter your conflict of interest statement in the “Confidential to Editor” section, and submit your "Accept" recommendation.

Reviewer #1: All comments have been addressed

Reviewer #2: All comments have been addressed

2. Does this manuscript meet PLOS Global Public Health’s publication criteria? Is the manuscript technically sound, and do the data support the conclusions? The manuscript must describe methodologically and ethically rigorous research with conclusions that are appropriately drawn based on the data presented.

Reviewer #1: Yes

Reviewer #2: Yes

3. Has the statistical analysis been performed appropriately and rigorously?

Reviewer #1: Yes

Reviewer #2: N/A

4. Have the authors made all data underlying the findings in their manuscript fully available (please refer to the Data Availability Statement at the start of the manuscript PDF file)?

Reviewer #1: Yes

Reviewer #2: No

5. Is the manuscript presented in an intelligible fashion and written in standard English?

Reviewer #1: Yes

Reviewer #2: Yes

6. Review Comments to the Author

Reviewer #1: (No Response)

Reviewer #2: (No Response)

7. PLOS authors have the option to publish the peer review history of their article (what does this mean?). If published, this will include your full peer review and any attached files.

**Do you want your identity to be public for this peer review?** For information about this choice, including consent withdrawal, please see our Privacy Policy.

Reviewer #1: No

Reviewer #2: **Yes: **Nolita Dolcy Saldanha
